

# Quenched dynamics and pattern formation in clean and disordered Bogoliubov-de Gennes superconductors

**Bo Fan⋆ and Antonio M. García-García†**

Shanghai Center for Complex Physics, School of Physics and Astronomy,
Shanghai Jiao Tong University, Shanghai 200240, China

⋆ bo.fan@sjtu.edu.cn , † amgg@sjtu.edu.cn

## Abstract

We study the quench dynamics of a two dimensional superconductor in a square lattice of size up to $200 \times 200$ employing the self-consistent time dependent Bogoliubov-de Gennes (BdG) formalism. In the clean limit, the dynamics of the order parameter for short times, characterized by a fast exponential growth and an oscillatory pattern, agrees with the Bardeen-Cooper-Schrieffer (BCS) prediction. However, unlike BCS, we observe for longer times a universal exponential decay of these time oscillations. We show explicitly that the origin of this exponential decay is the full emergence of spatial inhomogeneities of the order parameter characterized by the exponential growth of its variance. The addition of a weak disorder does not alter these results qualitatively. In this region, the spatial inhomogeneities rapidly develop into an intricate spatial structure consisting of ordered fragmented stripes in perpendicular directions where the order parameter is heavily suppressed. As the disorder strength increases, the fragmented stripes gradually turn into a square lattice of approximately circular spatial regions where the condensate is heavily suppressed. A further increase of disorder leads to the deformation and ultimate destruction of this lattice. We show these emergent spatial patterns are sensitive to the underlying lattice structure. We explore suitable settings for the experimental confirmation of these findings.

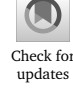

## Contents



## 1   Introduction

The spontaneous formation of patterns, defects and other spatial structures is a fascinating phenomenon rather ubiquitous in nature. It can be observed in various contexts, from the vortex pattern in superconducting thin films induced by a magnetic field [1] to the formation of spatial structures in a driven Bose-Hubbard model [2]. A particularly interesting phenomenon of defect formation, termed Kibble-Zurek mechanism [3,4], is the spontaneous generation of vortices as a result of a quench through a second-order phase transition which has been observed both experimentally [5–7] and numerically [8–10] in a wide variety of physical systems.

In many of these situations, far from equilibrium dynamics triggers spatial instabilities that eventually lead to pattern formation. In the context of superconductivity and superfluidity, although the study of nonequilibrium dynamics has received a lot of attention, the spontaneous generation of spatial structures, either due to the Kibble-Zurek mechanism or of different origin, has been modeled by employing phenomenological approaches such as the time dependent Ginzburg-Landau [1,11–13], the Gross-Pitaevskii equation [2,8] or applied holography [9,14] where the dynamics of the superconductor is mapped onto that of a gravitational system.

It is expected that these phenomenological approaches will be qualitatively correct close to a second order phase transition. However, the full non-linear structure of the time dependent Bogoliubov-deGennes (BdG) equations [15,16], due to the self-consistent condition verified by the order parameter, is fully necessary for the quantitative description of the out of equilibrium dynamics of a superconductor. A simpler problem, the quench dynamics of a Bardeen-Cooper-Schrieffer (BCS) superconductor [17], first investigated in Ref. [18], has received a lot of recent attention [19–24]. The conclusion of these studies is that details of the dynamics of the order parameter amplitude depend on both the initial state and the quench protocol, though a generic feature is the existence of oscillations in time. For an initial state above the critical temperature, it has been argued [19,20,25] that these oscillations do not decay with time, unless collision effects beyond BCS are taken into account. By contrast quenches in the coupling constant within the superconducting state [20–22] lead to oscillations whose amplitude typically decays either exponentially or as a power-law.

A perturbative analytic treatment of the dynamics using the full time dependent BdG formalism, that accounts for spatial inhomogeneities, showed that eventually the order parameter develops a simple oscillating spatial structure with a typical length directly related to the superconducting coherence length [26]. This perturbative treatment cannot take into account the full non-linear nature of the time evolution so the validity of these results is only assured for relatively short time scales where these non-linear effects are small. A very similar pattern of spatial oscillations has been reported in the quench dynamics of the order parameter of a holographic superconductor [14]. More recently [27], the spontaneous formation of an intricate spatial pattern resulting from the quench dynamic has been observed in the context of charge density waves

Despite this considerable progress, the dynamics of a superconductor after a quench, especially the nature of the emergent spatial structures, is still an open problem. We note that this is mostly due to technical challenges resulting from the combination of the non-linearity induced by the self-consistent condition and the requirement of sufficiently large system sizes in order to account for the emergent spatial structure from the quench dynamics. Moreover, it is necessary to consider at least a two dimensional superconductor because fluctuations in one dimension, even at low temperature, are too large to employ a mean field formalism.

In this paper, we address this problem by studying the quench dynamics of a two dimensional superconductor by the full self-consistent time-dependent BdG formalism [28–30] in a $200 \times 200$ square lattice that enables us to investigate in detail complex spatial patterns. We have found that for no disorder and short times, in agreement with the BCS results, the order parameter first grows exponentially and then has an oscillatory behavior. However, for longer times, time oscillations in the order parameter are suppressed exponentially independently of the quench protocol. The precursor of this behavior is the emergence of spatial inhomogeneities, beyond the reach of the BCS formalism, characterized by an exponential growth of the variance in space of the order parameter which ultimately results in the appearance of short stripes in the horizontal and vertical directions where the order parameter is heavily suppressed. We believe our results are largely universal as these spatial patterns occur well after the quench ends. The addition of a weak disordered potential, modeling impurities which are ubiquitous in experiments, does not change the above results qualitatively. As disorder increases, the mentioned fragmented stripes gradually morphs into a square lattice of *fake* vortices, namely, approximately circular regions where the amplitude of the order parameter is very small but with a trivial phase. Finally, as the insulating transition is approached, the lattice symmetry is eventually lost though the repulsion between *fake* vortices persists. In the next section, we introduce the model and the computation scheme.

## 2 The model

In order to study the time evolution of a two dimensional superconductor after a temperature quench, we employ the mean field time dependent BdG equations [28, 29, 31–34], which are given by

$$\begin{pmatrix} \hat{K} & \hat{\Delta}(\mathbf{r}_i, t) \\ \hat{\Delta}^*(\mathbf{r}_i, t) & -\hat{K}^* \end{pmatrix} \begin{pmatrix} u_n(\mathbf{r}_i, t) \\ v_n(\mathbf{r}_i, t) \end{pmatrix} = i\hbar \frac{\partial}{\partial t} \begin{pmatrix} u_n(\mathbf{r}_i, t) \\ v_n(\mathbf{r}_i, t) \end{pmatrix}, \tag{1}$$

where

$$\hat{K} u_n(\mathbf{r}_i) = -t_{i,i+\delta} \sum_\delta u_n(\mathbf{r}_{i+\delta}) + (V_i - \mu) u_n(\mathbf{r}_i), \tag{2}$$

$\delta$ stands for the nearest neighbors sites, and $t_{i,i+\delta}$ is the hopping energy between the nearest neighbors sites and we set $t_{i,i+\delta} = 1$ for simplicity. The onsite random potential $V_i$ is uniformly

distributed between $[-V, V]$, where $V$ is the strength of the disordered potential. $\mu$ is the chemical potential. This two parameters are in units of the hopping energy $t_{i,i+\delta}$.

To simulate the dynamical evolution, we solve the above equations by using the fourth order Runge Kutta algorithm [29, 35] with a sufficiently small time step $dt$ that ensure the convergence of the dynamics. To be more specific, we use $dt = 0.1/\Delta_0$ for system size $N = 200 \times 200$, and $dt = 0.01/\Delta_0$ for other smaller sizes where $\Delta_0$ is the value of the order parameter in the clean limit at zero temperature. The occupation number is assumed to satisfy the Fermi-Dirac distribution $f(E_n) = [1 + \exp(E_n/k_B T)]^{-1}$ at each time-step during the dynamical process, where $T$ is temperature and $k_B = 1$ is the Boltzmann constant. The time dependent order parameter is then defined as,

$$\Delta(\mathbf{r}_i, t) = |U| \sum_n u_n(\mathbf{r}_i, t) v_n^*(\mathbf{r}_i, t)[1 - 2f(E_n)], \tag{3}$$

where $U$ is the strength of the on-site, phonon induced, attractive electron-electron interaction, leading to the superconducting state. The time dependent local density is given by

$$n(\mathbf{r}_i, t) = 2 \sum_n [|u_n(\mathbf{r}_i, t)|^2 f(E_n) + |v_n(\mathbf{r}_i, t)|^2 (1 - f(E_n))], \tag{4}$$

For numerical convenience, we start from the equilibrium state at temperature $T_i > T_c$, namely, a vanishing order parameter which can be obtained from the exact solution of the BdG equations [36–38]. We note that due to the maximum numerical accuracy that our calculation can reach, even in the clean limit, the numerical error is of order $10^{-16}$. This numerical error induces a very weak spatial dependence in the initial state even without disorder. This numerical error is therefore the seed for the later emergence of spatial patterns if no disorder is present. We stress that by no means these spatial patterns are a numerical artifact. Physically, this seed has its origin in small thermal and quantum fluctuations that we are neglecting in mean field formalism that we employ but are always present in experiments. The details of the emergent spatial inhomogeneities induced by the quenched dynamics are largely independent on the origin of the seed.

Indeed, we also checked the quenched dynamics starting with a random but normalized initial state, in a smaller system size, leads to qualitatively similar results. We employ periodic boundary conditions to minimize finite size effects. Time evolution is induced by lowering the temperature $T(t)$ from $T_i > T_c$ to $T_f = 0.1T_c$ using the following linear quench protocol [9, 39],

$$T(t) = \begin{cases} T_i - \tau_Q t, & t_i \le t \le t_f \\ T_f, & t > t_f \end{cases}, \tag{5}$$

where $\tau_Q$, the slope of the quench, which characterizes the quench duration, has units $t_{i,i+\delta}^2/(\hbar k_B)$, $t_i = 0$ is the starting time of the quench, and $t_f = (T_i - T_f)/\tau_Q$ is the quench ending time corresponding to the final temperature $T_f$. In our study, we let $T_i = 1.2T_c$. We mostly focus on fast quenches leading to a non-adiabatic time evolution, so we set $\tau_Q = 50$. Since we are mostly interested in the generation of stable spatial patterns by the dynamics which occurs for relatively long time scales after the quench stops, we expect our results to be largely independent on the quench protocol. We quenched both the temperature and coupling constant with different quench speeds, and obtained qualitatively similar patterns, indicating that the quench dynamics is rather universal. More specifically, for zero disorder, we have checked, see Appendix B and C, that a quench in the coupling constant leads to qualitatively similar results for sufficiently long times. Moreover, since we aim to compare with the BCS dynamics for short times, our quench results in a superconducting state which for no disorder is still spatially homogeneous in the $T \lesssim T_c$ region.

# 3 Quench dynamics of the order parameter: initial exponential growth, time oscillations and its eventual suppression

We now proceed to study the dynamics of the condensate amplitude triggered by lowering the temperature of the system from $T_i > T_c$ to $T_f < T_c$ using the quench protocol Eq. (5). Since one of our main goals is the modeling of stable spatial patterns of the order parameter in the long time limit [26], we use the self-consistent time dependent BdG equations introduced earlier which results in an initial spatially homogeneous evolution of the order parameter. We shall see that a weak disordered potential does not change this picture substantially. This is indeed a welcome feature as another aim of the paper is to compare for short times our results with previous theoretical predictions using the simpler BCS approach that cannot account for spatial inhomogeneities.

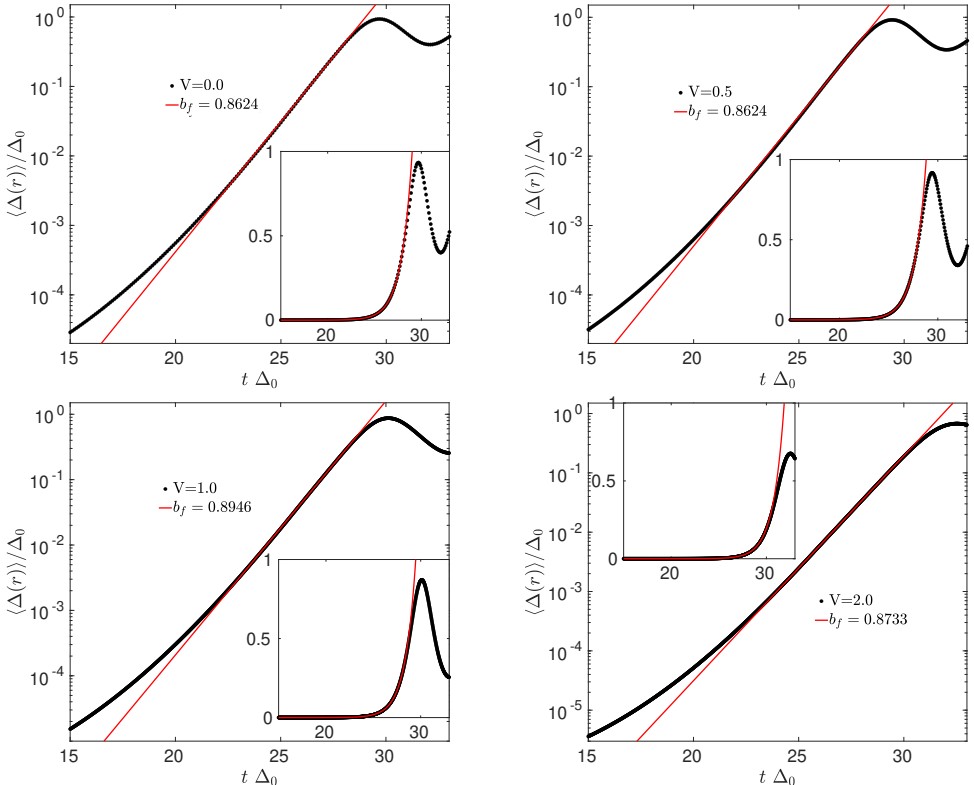

Figure 1: The dynamics of the spatially averaged order parameter $\langle\Delta(\boldsymbol{r})\rangle$ (black dot), normalized by $\Delta_0 \sim 0.83$, together with an exponential fitting $f(t) = a_f \exp(b_f t)$ (red line) at short times corresponding to temperatures slightly below the critical one that marks the transition into the superconducting state. The other parameters are system size $N = 200 \times 200$, the coupling constant $U = -3$ and the chemical potential is $\mu = -0.34$ corresponding to a mean charge density $\langle n \rangle \simeq 0.875$.

We first study the initial growth of the condensate as the system enters the superconducting phase. We observe that the amplitude of the order parameter increases rapidly from zero as the system enters the superconducting phase by lowering the temperature $T < T_c$. A careful fitting of the numerical results, see Appendix A, indicates that the growth of the order parameter is exponential in this region. This confirms that the quench is fast enough to induce a highly non-adiabatic time evolution. An exponential growth is also observed, see Figure 1, in a relatively broad range of disorder strengths with a growth rate, $0.86 \sim 0.89$, that is not very sensitive to disorder and it is also similar (0.86) to that found in the clean limit.

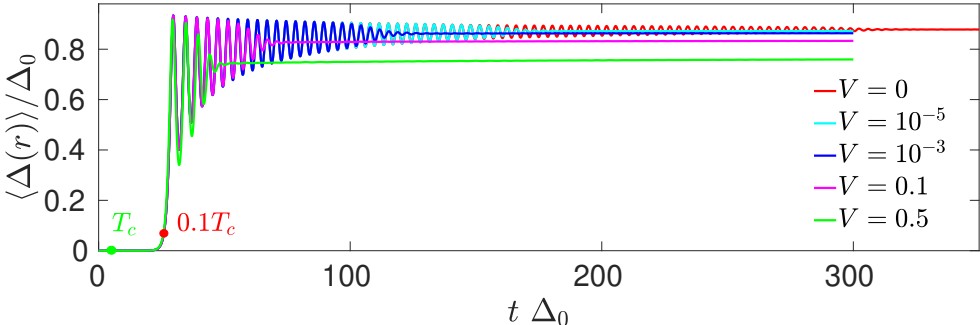

Figure 2: Time evolution, after spatial average, of the order parameter $\langle \Delta(r) \rangle$ in the presence of a random potential for different values of the disorder strength $V$. The other parameters are the same as in Figure 1.

The initial exponential growth of the spatial averaged order parameter amplitude, obtained from the solution of the BdG equations Eq. (1), is followed, see Figure 2, by relatively simple oscillations in time. At this early stage, we did not yet observe spatial inhomogeneity in the clean limit, or even in the presence of a weak disordered potential $V \leq 0.1$. The time evolution of the order parameter within the simple BCS formalism [19–22, 25], that by design neglects spatial inhomogeneities, also shows oscillations in time whose details depend on both the initial state and the quench protocol. More specifically, for an initial state of the order parameter characterized by uncorrelated phases in momentum space [25] or with a very small initial value [20], which simulates the system above the critical temperature $T_c$, the lowering of the temperature below $T_c$ induces undamped periodic oscillations in the amplitude of the order parameter. By contrast, for a quench in the coupling constant at zero temperature, and therefore inside the superconducting phase [20–22], the amplitude of the order parameter oscillations can decay either as a power-law or exponential way, or not decay at all, depending on the values of the initial and final coupling constants. As expected, we have found excellent agreement with the BCS prediction for the protocols that we have tested explicitly, see Appendix C. This is not surprising as BCS theory and BdG theory should agree in the limit in which the order parameter is spatially homogeneous.

The dynamics becomes more interesting for longer times. Results depicted in Figure 2 indicate that this simple dynamical regime ends rather abruptly due to the sharp exponential suppression of the amplitude of these oscillations, which does not occur in the BCS dynamics [19–22, 25], see also Appendix C. The time scale of this suppression is sensitive to the addition of a weak random potential. For a stronger disorder still deep in the metallic phase, the oscillations are almost fully suppressed after a few periods. We show next, by employing the mentioned time dependent BdG formalism, that the origin of that exponential suppression lies in the development, even for no disorder, of spatial inhomogeneities in the order parameter.

In order to carry out a more quantitative analysis of the decay of the amplitude of the order parameter, we define $\delta \Delta = (\langle \Delta(r) \rangle_{\text{peak}} - \langle \Delta(r) \rangle_{\text{valley}})/\Delta_0$, see Figure 3, where peak and valley refer to consecutive local maxima and minima of the oscillations in time of the order parameter. For sufficiently short times, where the order parameter is spatially homogeneous, the reduction of the amplitude $\delta \Delta$ is consistent with a power-law decay. A fit with a power-law decaying function $\sim t^{-\gamma_p}$ yields $\gamma_p \sim 1.4$ for weak or no disorder which illustrates that this slow decay is not related to the presence of a random potential.

The power-law decay is followed by a much faster exponential decay $\sim \exp(-\gamma_e t)$ even in the clean limit. As is expected, $\gamma_e$ becomes larger as disorder increases. Even for a relatively weak disorder, $V = 0.1$, it is already about two times larger that in the clean limit. Qualitatively, the dependence of the crossover time between power-law and exponential decay on

the disorder strength $V \ll 1$ seems to be rather weak which reinforces the idea that disorder does not play a leading role in this phenomenon. We investigate in more detail this region of exponential suppression in the next section.

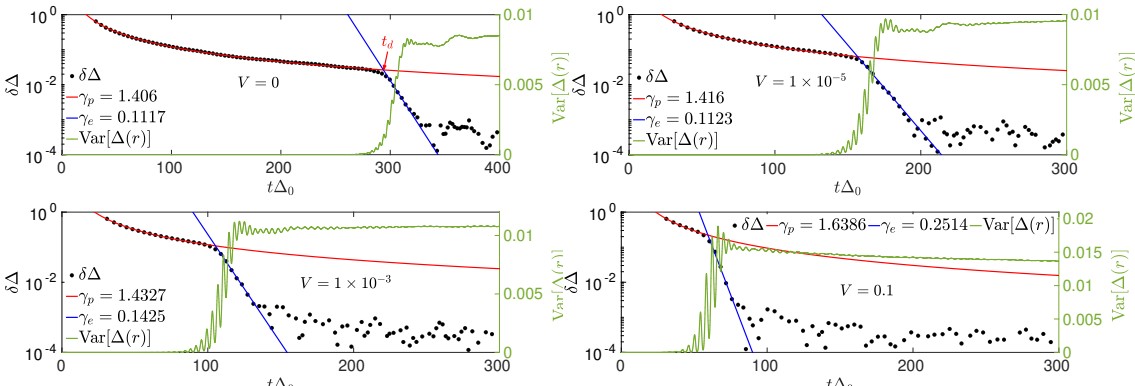

Figure 3: Left y axis: the amplitude of the time oscillation $\delta\Delta$ (black dots) of the order parameter in the presence of different disorder strengths $V$ and the corresponding power-law $y \propto t^{-\gamma_p}$ (red line) and exponential fittings $y \propto \exp(-\gamma_e t)$ (blue line). Right y axis: the variance of the order parameter $\text{Var}[\Delta(r)]$. We combine both results using a double y axis plot in order to show explicitly that the exponential suppression of time oscillations is induced by the exponential growth of spatial inhomogeneities. Only when the spatial inhomogeneities become sufficiently large, because of the exponential growth of the variance, the exponential suppression of the time oscillations occurs.

## 4 Exponential growth of emergent spatial inhomogeneities and exponential suppression of time oscillations

We now carry out a comprehensive study of the full form of the exponential decay of the order parameter oscillations in time aiming to relate this exponential suppression with the emergence of spatial inhomogeneities in the order parameter even in the absence of disorder.

As a first step, we employ a simple oscillatory function with an exponential decay of the amplitude [19, 26] to fit the time dependent spatial averaged order parameter obtained from the BdG formalism,

$$\Delta(t) = \Delta_f - A(\Delta_f + C + \cos(\omega(t - t_0)\Delta_0))/\exp(\gamma(t - t_0)\Delta_0), \tag{6}$$

where $\Delta_0 \sim 0.83$ is the value of the order parameter in the clean limit at zero temperature, and $\Delta_f$ is the order parameter in the final equilibrium state after the quench. The four fitting parameters are $\gamma$, the decay ratio of the amplitude, $A$, $C$ and $\omega$. As is shown in Figure 4, for times right after the crossover to an exponential decay in $\delta\Delta$ (see Figure 3) , we find a very good agreement with the BdG results in the clean and weak disorder limit. We note that in the BCS approach, the value of $\omega$ is sensitive to the initial state and the quench protocol [19, 20, 40]. In our case, the fitting yields $\omega \sim 0.3\Delta_0$ which is in the same ballpark as the BCS prediction [40] for an initial state characterized by a very small order parameter with respect to $\Delta_0$. In any case, we do not expect quantitative agreement because this frequency may also be affected by the emergence of spatial inhomogeneities.

For earlier times, as expected, the decay of time oscillations is much slower so the fitting is much worse. Moreover, we find that this exponential decay, even for no disorder, seems to

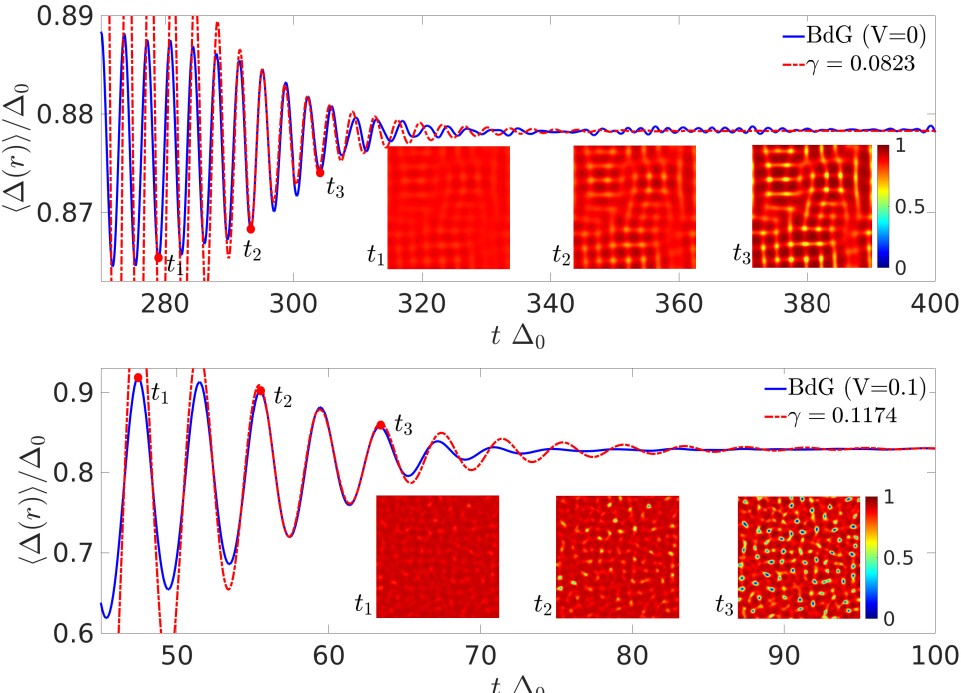

Figure 4: The time dependence of the spatially averaged order parameter $\langle \Delta(\boldsymbol{r}) \rangle$ (blue line) in the clean $V = 0$ and weak disorder regions $V = 0.1$. The red dash-dot line is the best fitting, by Eq. (6), in the region of exponential decay of the amplitude of the time oscillations of the order parameter that follows the region of slower power-law decay. As is observed in the inset plots of the spatial distribution of the order parameter at times $t_1, t_2, t_3$, this time scale is related to the formation of spatial inhomogeneities. As was expected, the fitting parameter $\gamma$ which characterizes the exponential decay increases with increasing disorder. The system size is $N = 200 \times 200$, the coupling constant $U = -3$, and the chemical potential $\mu = -0.34$.

be closely related with the emergence of spatial inhomogeneities in the order parameter. This can be seen from the similarity between the time scale $t_2$ in which the exponential suppression of oscillations occurs and the time scale $t_3$ in which spatial inhomogeneities, already existent for $t_2$, become substantial, see the insets of Figure 4. In other words, the emergence of spatial inhomogeneities eventually, namely, when they are large enough, triggers the exponential decay of oscillations in time that terminates approximately when the spatial inhomogeneities are fully formed.

In order to establish a more quantitative relation between spatial inhomogeneities and the exponential decay of time oscillations, we compute the variance of the order parameter $\mathrm{Var}|\Delta(\boldsymbol{r})| = \langle \Delta^2(\boldsymbol{r}) \rangle - \langle \Delta(\boldsymbol{r}) \rangle^2$ as a function of time in the clean limit and in the presence of weak disorder $V \leq 0.1$. Results depicted in Figure 5 show a region of intermediate times where the variance grows exponentially. We define $t_e$ as the time in which this exponential growth terminates because the spatial patterns are completely developed, see Figure 5a.

Another interesting feature of the time dependence of the spatial variance is the observation of a period of no growth, only for no disorder $V = 0$, right after the initial growth of the condensate. This feature strongly suggests that the later exponential growth is independent of the quench protocol. For $V \neq 0$, the situation is different. The early exponential growth in time of the amplitude of the order parameter is followed by the exponential growth of the variance which indicates that the seeds of spatial inhomogeneities due to disorder are amplified exponentially by the dynamics of the BdG superconductor. The flat behavior of the variance

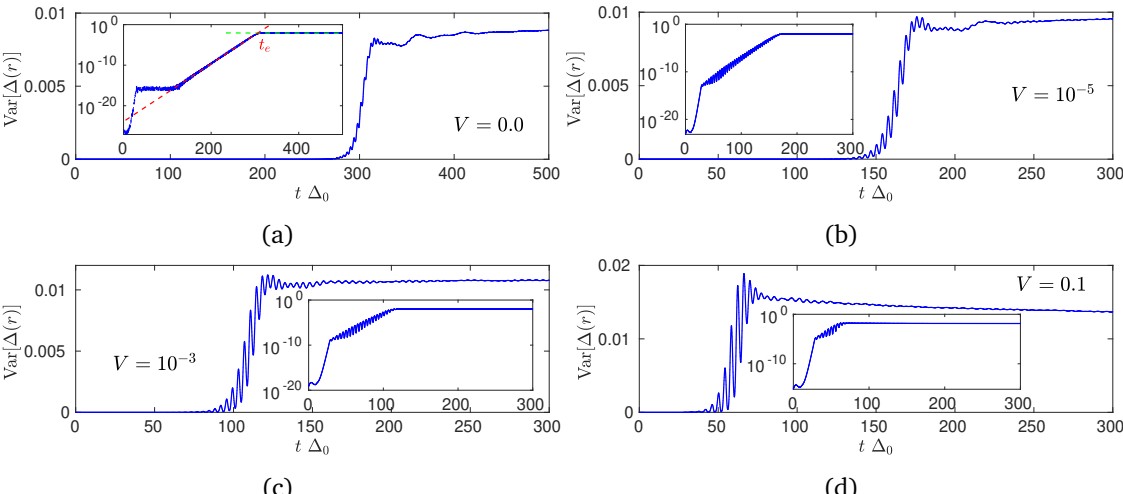

Figure 5: Variance of the spatial inhomogeneities of the amplitude of the order parameter $\Delta(\boldsymbol{r}, t)$ for different disorder strengths $V$. The starting of the exponential growth observed for intermediate times occurs much earlier than the beginning of the exponential suppression of the oscillations of the averaged order parameter.

for later times, after the exponential growth, confirms the earlier claim that the end of the time oscillations is related to the approach to a quasi equilibrium state where the change with time of the order parameter is heavily suppressed.

We have now all ingredients to compare explicitly to what extent the exponential growth of the spatial inhomogeneities and the crossover between power-law and exponential decay of oscillations in time are closely related. For that purpose, we depict in Figure 3, back to back, the plots of $\delta\Delta(t)$, that characterizes the amplitude of time oscillations, and $\mathrm{Var}[\Delta(r)]$, representing the exponential growth of the variance of the spatial inhomogeneities. For both, the clean and the weak disordered case, $V \leq 0.1$, the crossover to an exponential decay of the oscillations precisely occurs when the exponential growth of the variance of spatial fluctuations is close to its termination, namely, when it has reached a value sufficiently large to have an impact on the quench dynamics. Therefore, $t_d$ and $t_e$ has a very similar dependence on disorder and $t_d$ is a bit larger than $t_e$ in all cases as the exponential suppression does not require a full development of spatial inhomogeneities described by $t_e$. These results fully confirm that the oscillations in time of the order parameter are eventually suppressed exponentially by the emergence of spatial inhomogeneities characterized by a variance that grows exponentially. This is therefore a quite robust feature of the non-adiabatic dynamics of BdG superconductors.

We turn to a quantitative investigation of the dependence on disorder of these results. For that purpose, we show in Figure 6, the dependence on the strength of disorder $V$ of $t_d$, the typical crossover time between power-law and exponential decay of $\delta\Delta(t)$, and $t_e$ the time at which the exponential growth of the variance terminates. We observe that in the weak disorder limit $V \leq 0.1$ of interest, both typical times have not only similar values but also a simple logarithmic dependence with the disorder strength, with a finite value for $V = 0$, confirming, at least for weak disorder, that the exponential decay of the order parameter time oscillations has its origin in spatial inhomogeneities induced by the quench dynamics with the disorder potential playing the secondary role, at least in this region, of shifting the development of these spatial pattern to earlier times. We note that the slightly larger value of $t_e$ is expected because the effect of the spatial inhomogeneities must be felt earlier, but not much earlier because the growth is exponential, than the time at which the spatial inhomogeneities are fully formed.

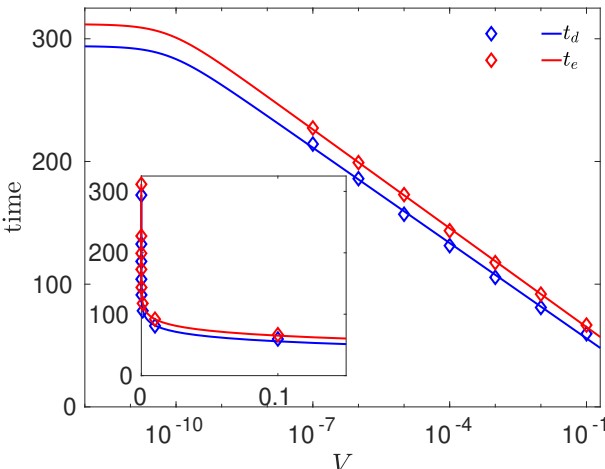

Figure 6: Typical crossover time $t_d$ (blue rhombs), see Figure 3a, between the power-law and exponential decay of the amplitude of the time oscillations of the order parameter as a function of the disorder strength $V$. Typical time $t_e$ (red rhombs) signaling the end of the exponential increase of the variance of the order parameter spatial inhomogeneities depicted in Figure 5a. The solid lines are fits using the function time = $a_d \log(b_d \times V + c_d)$. We not only observe a similar logarithmic dependence in both cases, with a finite value at $V = 0$, but $t_d$ and $t_e$ have similar values. Therefore, the exponential growth of the spatial inhomogeneities induces the late exponential decay of the amplitude of the time oscillations.

We now move to the quantitative description of the spatial patterns for sufficiently long times when the system reaches the final quasi equilibrium state.

## 5 Long times: spontaneous formation of spatial patterns resulting from the quench dynamics

The conclusion of the previous section is that, within the time dependent BdG formalism, time oscillations of the order parameter are exponentially suppressed for sufficiently long times. This is markedly different from the BCS result in which this specific exponential suppression does not occur, because the order parameter is spatially homogeneous.

In this section, we present a detailed description of those stable, spontaneously formed, spatial patterns, see Figure 7, together with its formation process, see Figure 8, as a function of the disorder strength. Videos of the full time evolution are available here.[1] We recall that for no disorder, spatial inhomogeneities start to appear when the time oscillations are significantly suppressed. Initially, see left column of Figure 8, they resemble soft periodic or quasi-periodic domains where the order parameter amplitude is substantially smaller than in the surroundings. For longer times, these domains become more pronounced and adopt the shape of relatively thin finite-size stripes, organized in most cases in a perpendicular fashion. These broken stripes have well defined centers where the suppression of the order parameter is even stronger. This seems to be an equilibrium or quasi-equilibrium state because we do not appreciate further changes even for much larger time scales. A small but finite disorder does not change much the emergence of these spatial patterns.

---

[1]Videos showing the time evolution of the order parameter after a temperature quench for different disorder strengths V are found in [41]

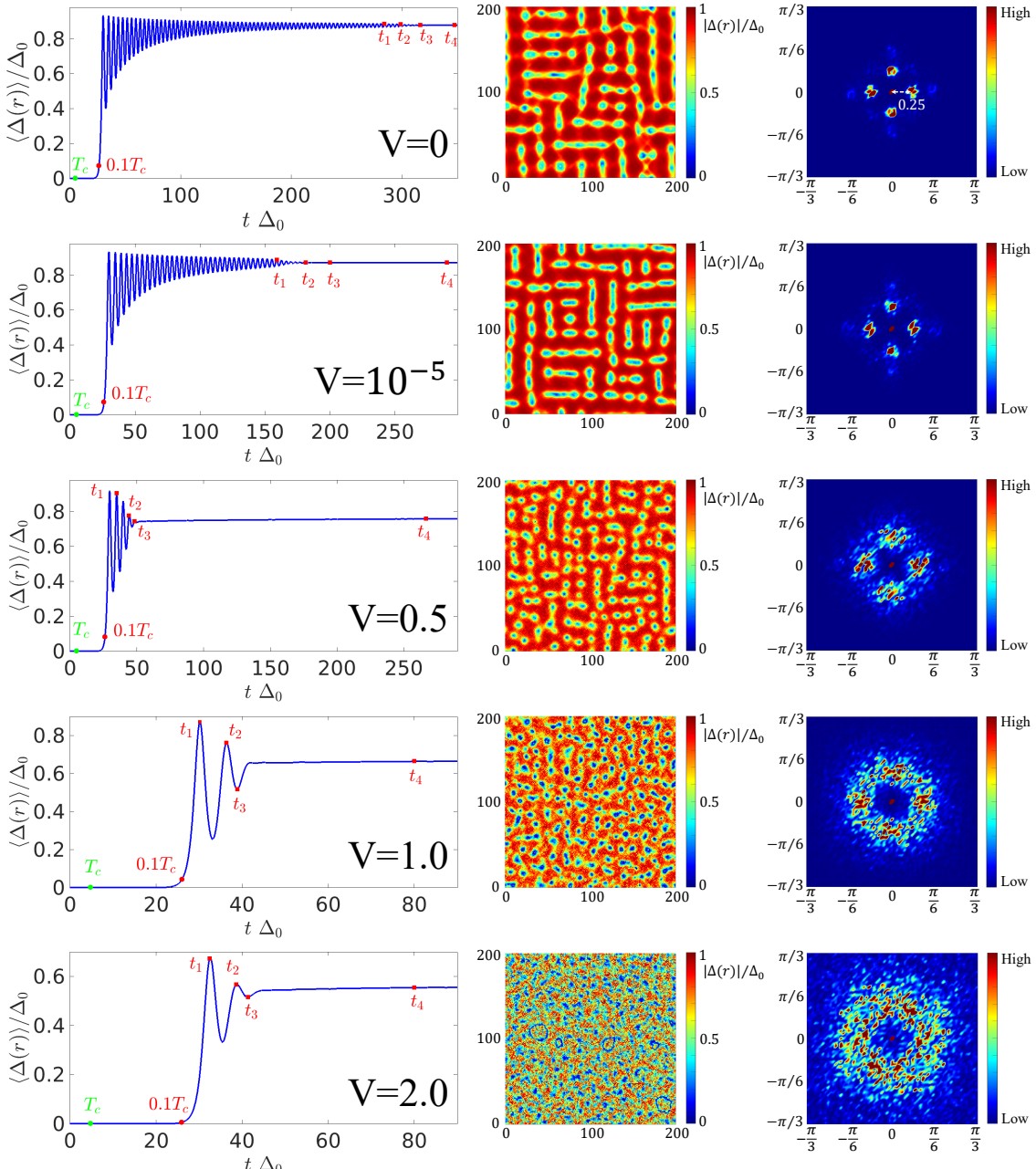

Figure 7: Left column: the time dependence of the spatial averaged order parameter $\langle \Delta(\boldsymbol{r}) \rangle$ for different disorder strengths $V$. Central column: spatial distribution of the order parameter at time $t_4$, see left column, corresponding with a quasi-equilibrium state with intricate spatial patterns and a very weak time dependence. Right column: structure factor Eq. (7) that reveals patterns in the spatial distribution of the order parameter at time $t_4$. The parameters are: system size $N = 200 \times 200$, coupling constant $U = -3$ and chemical potential $\mu = -0.34$.

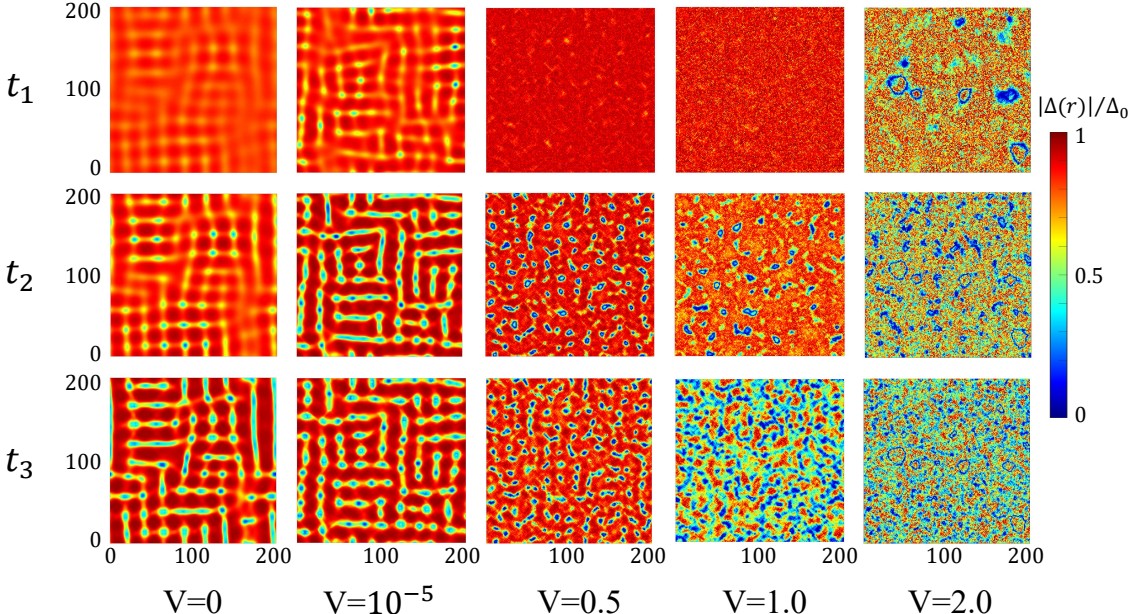

Figure 8: The spatial distribution of the order parameter $\Delta(\boldsymbol{r})$ at times $t_1$, $t_2$ and $t_3$, defined in Figure 7, corresponding to different stages of the time evolution before the full quasi-equilibrium state ($t_4$), and for different values of the disorder strength $V$.

In order to have a more quantitative description of this spatial structure, we have computed the averaged gap correlation function $\langle \Delta(\boldsymbol{r})\Delta(0)\rangle$. Results depicted in Figure 9 show oscillations with a typical length $\ell_p \sim 12.5$ in the clean limit, which is much larger than the superconducting coherence length $\xi_0 \sim 1$ at thermal equilibrium. This is a strong indication that the pattern of spatial inhomogeneities, especially its periodicity, is due to the quench dynamics.

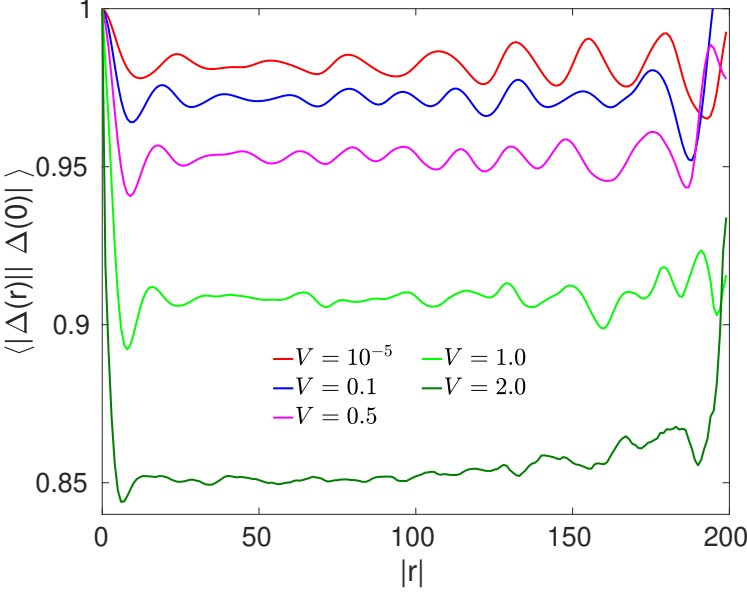

Figure 9: The static order parameter correlation function at time $t_4$, defined in Figure 7, normalized by the spatial average of the square of the order parameter.

As a further probe of the periodic nature of the spatial inhomogeneities, we compute the structure factor,

$$S(\boldsymbol{q}) = \sum_{ij} \Delta(\boldsymbol{r}_i)\Delta(\boldsymbol{r}_j)\exp(i\boldsymbol{q}(\boldsymbol{r}_i - \boldsymbol{r}_j))/\sum_i \Delta^2(\boldsymbol{r}_i)\,, \qquad (7)$$

of the distribution of the order parameter for long times where the system seems to have reached a quasi equilibrium state. We termed quasi-equilibrium state for two reasons, there is still some residual time dependence and also the resulting state is very different from that corresponding to the solution of the static BdG equations. As is shown in Figure 7, the spatial patterns resulting from the out of equilibrium dynamics have a square crystal-like structure. We have checked that the structure of this spatial pattern depends on the underlying lattice structure. In the Appendix B, we obtain a Hexagonal pattern in the structure factor for a triangular lattice system. Moreover, according to the Bragg pattern depicted in the right column of Figure 7, the distance from the center to the peaks is around 0.25 in momentum space which in real space corresponds to a typical length of the strips, or the fake vortex lattice, of about $\ell_p \sim \pi/0.25 = 12.56$. This is consistent with the previous findings for the order parameter spatial correlation function in Figure 9. The addition of a weak disorder potential does not modify substantially these results.

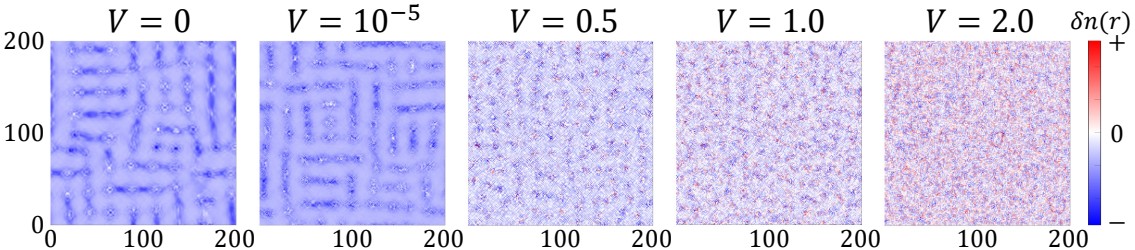

Figure 10: The spatial variation of the charge density $\delta n(\boldsymbol{r}) = n(\boldsymbol{r},t) - n(\boldsymbol{r},t=0)$ at time $t_4$, defined in Figure 7. The spatial pattern of the charge density matches that of the superconducting order parameter, especially when disorder is not very strong.

Results depicted in Figure 10 for the charge density variation $\delta n(\boldsymbol{r})$ further confirm the existence of spatial patterns consistent with the one found for the spatial structure of the order parameter. Therefore, the observed stripe-like domains where the order parameter is suppressed cannot be attributed to quantum coherence effects [37,42,43], but are rather related to modulations of the charge density caused by the strong suppression of time oscillations.

## 5.1 From fragmented stripes to *fake* vortices

As disorder strength increases, but still deep in the metallic region, we observe a similar periodic pattern that does not yet seem much affected by disorder. Although the periodicity is robust, see Figure 7, the shape of the domains where the order parameter is suppressed undergoes a gradual change. The stripes becomes more fragmented until they become completely disconnected. However, a rather strict periodicity of these patterns is still observed even for substantially larger disorder $V \leq 0.5$. The regions where the order parameter is suppressed are now circular and with a typical length that is much larger than the superconducting coherence length. As we said earlier, the phase of the order parameter is fixed in this region, so the periodic suppression of the order parameter is not related to non-trivial topological properties. For that reason, we have termed this spatial pattern *fake* vortex lattice. Both, the correlation function $\langle \Delta(\boldsymbol{r})\Delta(0) \rangle$, see Figure 9, and the structure factor of the spatial distribution of the superconducting order parameter, see Figure 7, confirm this lattice structure of the spatial patterns.

For $0.5 < V < 2$, disorder effects become gradually more pronounced. The periodic pattern of the fake vortex is gradually deformed though clear vortex repulsion is still observed even when $V = 1.0$. The latter is confirmed by an explicit calculation of the structure form factor, see Figure 7, that still shows a clear circular pattern. This is also confirmed in the order parameter correlation function $\langle \Delta(\mathbf{r})\Delta(0) \rangle$, see Figure 9, where the vestiges of a lattice structure resulting in vortex repulsion show up as a valley at $|\mathbf{r}| = 8$ and a peak at $|\mathbf{r}| = 16$ for a relatively large strength $V = 1.0$ of the random potential.

In the strong disorder limit, close or around the transition $V \sim 2$, the spatial distribution of the order parameter is mostly controlled by disorder. It becomes highly inhomogeneous, has no specific periodic pattern but it seems that some vortex-like structures remain. For instance, the gap correlation function $\langle \Delta(\mathbf{r})\Delta(0) \rangle$, see Figure 9, only has short correlations in space and the Fourier transform has no Bragg peaks signaling no periodic pattern in space.

# 6  Discussion

A natural question to ask is the origin of the crossover from stripes to *fake* vortices that we observe as the strength of disorder is increased. Since disorder suppresses finite size effects, we cannot rule out that the so called *fake* vortices, that arise for stronger disorder, should still be observed for weak or no disorder, instead of the fragmented stripes, if it were possible to reach much larger lattice sizes. Unfortunately, substantially larger sizes are currently beyond the reach of our computing capabilities.

Another important issue is the dependence of the results to the strength of the electron-phonon coupling. We have set the coupling constant in the strong coupling region $U = -3$, and set the Debye energy to the full band of the energy spectrum, in order to be able to explore quantitatively the rich spatial structure of the order parameter resulting from the out of equilibrium dynamics whose typical length is around $\ell_p \sim 12.56$, which is much larger than the superconducting coherence length. As is expected, this is not fully consistent with the weak-coupling analytic prediction [26] that this typical length should be the superconducting coherence length. Results for an even stronger coupling constant $U = -5$, presented in the Appendix E, show similar spatial patterns but, as expected, the typical length of the patterns is shorter. A more quantitative understanding on the precise nature of this length scale would require a more systematic, and therefore numerically costly, analysis of its dependence on the quenched dynamics, for instance by varying the coupling constant, which is beyond the scope of the paper.

More specifically, it may be interesting to explore the weak coupling region $|U| \leq 1$ where quantum coherence effects induce multifractal-like features for intermediate to strong disorder. However, we anticipate that typically strong quantum coherence effects such as multifractality will likely suppress the spatial patterns induced by the quench dynamics even for a relatively weak disorder strength $V \leq 0.5$. This bring us to the issue of the experimental confirmation of these results. The strong coupling region we have explored in this paper is more amenable for experiments with Bose-Einstein condensate at very low temperature where the dynamics is triggered by a change in the coupling constant which is feasible to carry out in this setting. Disorder-like effects in this setting can be modeled by quasi-periodic optical lattice configurations.

We note that in superconducting materials, a quench protocol based on the change of coupling constant is not in principle possible but even a controlled quench in temperature is challenging. This is why some experiments opted to induce out of equilibrium dynamics by bombarding the sample with photons leading to heating and subsequent cooling [44]. However, the theoretical modeling of such systems is well beyond the mean-field approach, and

relatively simple quench protocols, that we are considering here. Having said that, we believe that a mean field approach is enough for a description of the physics behind the emergence of large spatial inhomogeneities of the condensate even in two dimensions. We note that quantum and thermal fluctuations beyond the mean field formalism, if sufficiently small, which is the case for low temperatures, large sizes and not very strong electron-electron interactions, induce very small spatial inhomogeneities in the order parameter that act as seed for the later emergence of spatial patterns. As mentioned earlier, in our numerical formalism, the seed is a consequence of the finite accuracy of our numerical calculation so, indirectly, we are taking these effects into account. Larger effects such as the Berezinskii–Kosterlitz–Thouless transition [45, 46] can be avoided with quenches ending at sufficiently low temperatures.

Finally, we address the dependence of the results on the quench speed. Results depicted in Appendix D for a much slower quench, points to a more nuanced picture. For sufficiently long times, spatial inhomogeneities are clearly observed but they do not have a periodic pattern. Large domains with similar values of the order parameter are separated by filamentary domain walls where the order parameter is highly suppressed. The domains walls becomes thinner for longer times but they persist in the range of times we can explore numerically. We therefore expect that a sufficiently slow quench will in principle lead to an essentially adiabatic dynamics where early time oscillations are suppressed and spatial patterns may not develop at all. We also note that sufficiently fast temperature quenches inducing far out of equilibrium effects are beyond the finite temperature mean-field formalism that we employ in the paper.

# 7  Conclusions

We have investigated the quenched dynamics of a BdG superconductor. In order to compare with previous results, which use the simpler, spatially homogeneous, BCS formalism, we have employed two quench protocols: an abrupt change in the coupling constant, see Appendix C and B, and a smooth linear drop in temperature starting above the critical temperature. For zero disorder and sufficiently long times, where the BCS approach ceases to be applicable, we have obtained similar results so the study of the role of disorder in the main text was carried out only for the second quench protocol.

For short times, we observe similar results as in the simpler BCS approach, the order parameter first grows exponentially and then oscillates in time with a pattern that depends on the quench protocol and the initial state. However, in contrast with previous BCS findings, the amplitude of these time oscillations eventually decreases exponentially in time because of the emergence of spatial inhomogeneities of the order parameter. We have characterized the emergence of these spatial inhomogeneities by the exponential growth of the variance in space of the order parameter and have shown that this exponential growth in space causes the exponential suppression of the oscillations in time. This feature cannot be accounted for in the BCS formalism and it is observed even in the limit of no disorder. A weak disordered potential does not change qualitatively these results.

For longer times, these spatial instabilities turn into rich spatial patterns. In the clean limit, or for sufficiently weak disorder, we observe ordered filamentary structures resembling finite size stripes in close to perpendicular directions, where the order parameter is heavily suppressed. The suppression of the order parameter in the central region of these broken stripes is much more pronounced. Those emergent spatial patterns depend only on the underlying lattice structure and not on the employed quench protocols. For stronger disorder, but still deep in the metallic region, the stripe-like structures turn into a square lattice of *fake* vortices, namely, the amplitude of the order parameter is heavily suppressed in circular regions whose typical length is much larger than the superconducting coherence length, but the phase has no

topological properties so no real vortex is formed. Larger sizes, beyond our current numerical capabilities, are needed to clarify whether the observed broken stripes for weak or no disorder become also a vortex lattice in the thermodynamic limit. A further increase of disorder leads to a gradual deformation of the lattice, though isolated fake vortices that repel each other are still clearly observed. Finally, close to the insulating transition, spatial inhomogeneities induced by disorder become also relevant leading to a quite intricate spatial structure that is characterized by a lack of vortex lattice symmetry but persistence of vortex repulsion. Although the time scale we can simulate numerically is limited, our results suggest that, neglecting the effect of collisions beyond the time dependent BdG formalism, these spatial structures correspond with a quasi-equilibrium state of the superconductor which is still very different from that corresponding with the solution of the static BdG equations. The observed spatial patterns are confirmed by a careful analysis in Fourier space based on the calculation of the structure factor. We have shown, by an explicit calculation with an underlying triangular lattice, that the emergent spatial patterns are sensitive to the underlying lattice structure which points to a rather rich spectrum of possible patterns of spatial inhomogeneities induced by the quenched dynamics.

It would be interesting to gain a more comprehensive understanding of the precise nature of the spatial patterns, especially in the so called stripe region, either by numerical or analytic techniques, in order to determine whether, for sufficiently large sizes, the spatial structures that emerge for intermediate times lead to a full checkerboard-like shape instead of the observed broken stripes. It would also be worthwhile to extend these results to p-wave and d-wave superconductors in order to explore its potential relevance in topological superconductivity and the physics of cuprates superconductors.

# Acknowledgements

We acknowledge Prof. Zi Cai for helpful and illuminating discussions.

**Funding information** We acknowledge support from the National Natural Science Foundation of China (NSFC): Individual Grant No. 12374138, Research Fund for International Senior Scientists No. 12350710180, National Key R&D Program of China (Project ID:2019YFA0308603), and China Postdoctoral Science Foundation (Grant numbers: 2023M732256, 2023T160409, GZB20230420).

# A The initial growth of the condensate

In this appendix, we study the initial growth of the superconducting order parameter when the temperature is just lowered below the critical temperature. For that purpose, we fit numerical results from the solution of the BdG equation with Gaussian, exponential and power-law test functions. The outcome of the fitting, see Figure 11, is that the order parameter increases exponentially during the early stages of the time evolution in the superconducting phase. This is consistent with previous results in phenomenological models for sufficiently fast quenches [9].

# B The dynamic pattern formation in the triangular lattice

In the main text, we have presented a comprehensive analysis of the square lattice, which lead to the observed square pattern of spatial inhomogeneities for sufficiently long times. In this

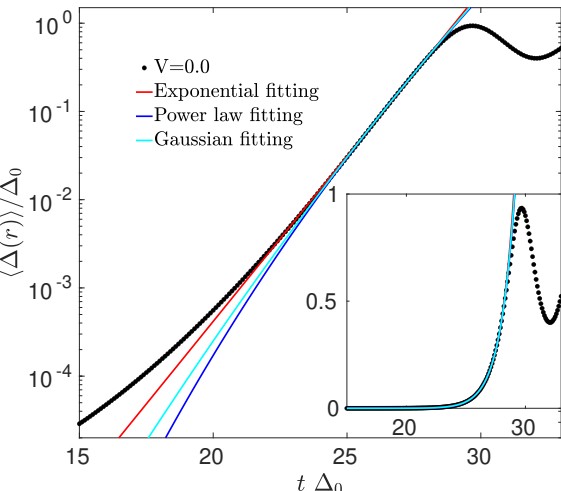

Figure 11: The dynamics of the spatially averaged order parameter $\langle \Delta(\boldsymbol{r}) \rangle$ (black dot) for times right after the system has entered in the superconducting phase. The results of the different fittings using Gaussian, power-law and exponential functions clearly indicate that the latter is the one closer to the numerical results.

appendix, we extend our investigation to a BdG superconductor with an underlying triangular, instead of square, lattice. A summary of the quenched dynamics is depicted in Figure 12. Although the pattern in real space may not be that evident, the corresponding structure factor analysis reveals a clear hexagonal pattern. By studying different lattice configurations, we gain valuable insights into the influence of the lattice geometry on the formation of spatial patterns induced by the out of equilibrium dynamics. Those results indicate that the underlying lattice symmetry plays a crucial role in the form of the emergent spatial patterns for sufficiently long times after the quench.

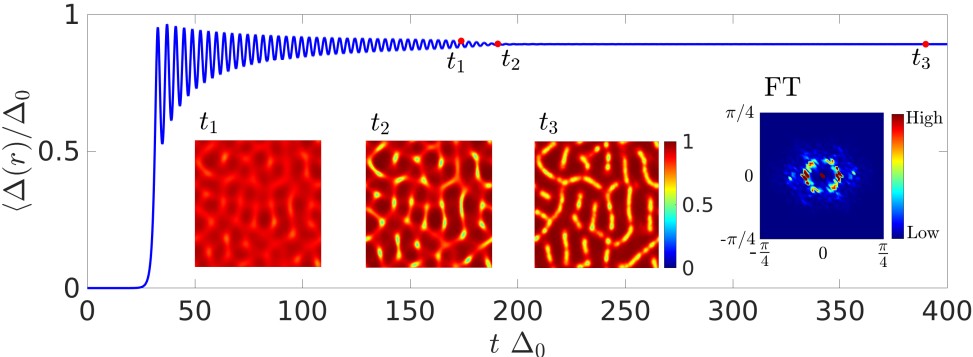

Figure 12: The time evolution of the spatial averaged order parameter of a BdG superconductor living in a two dimensional triangular lattice. The coupling constant $U = -4$ and the chemical potential $\mu = 0.108$ leading to $\Delta_0 = 1.098$. The insets show the spatial distribution of the order parameter at three representative times marked by red points and the corresponding structure factor at time $t_3$. The other parameters are system size $N = 200 \times 200$, disorder $V = 10^{-5}$, initial temperature $T_i = 1.2 T_c$, and final temperature $T_f = 0.1 T_c$ and $\tau_Q = 50$.

## C  Comparison between the dynamic BCS and BdG results in the sudden coupling quench case

In this appendix, we compare the quench dynamics using the BCS and BdG formalism. We cannot use the quench in temperature of the main manuscript because it is difficult to model it in the BCS formalism which cannot, at least in a fully self-consistent way, account for spatial inhomogeneities. Therefore, we compare the quench dynamics in BCS and BdG by using a quench protocol with an abrupt change in the coupling constant at zero temperature where the initial state is spatially homogeneous and therefore it is possible to model it with both approaches. Results depicted in Figure 13 show that in the beginning, when spatial dependence is not yet observable, the BCS and BdG quench dynamics is quantitatively very similar. However, at time $t\Delta_f$ around 400, we start observing the spatial patterns in the BdG results which triggers a sharp departure from the BCS prediction. Those results, together with those presented in the main text corresponding to the dynamics after lowering the temperature into the superconducting phase, show that the emergent spatial structure as a consequence of the quench dynamics is rather universal and independent on the quench protocol.

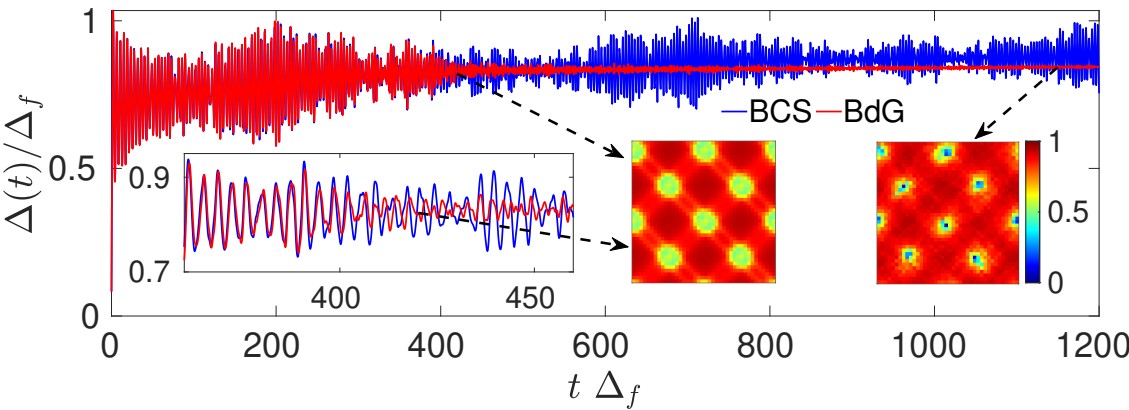

Figure 13: The time evolution of the order parameter $\Delta(t)/\Delta_0$ in the clean limit obtained with the BCS (blue) and BdG (red) formalisms. The insets show (left) the time range when the BdG result start deviating from the BCS one and (right) the spatial distribution of the order parameter at the time marked by the black arrow. The system is prepared in an initial state with coupling constant $|U| = 1$. The quench protocol consists in an abrupt change of the coupling constant from $|U| = 1$ to $|U| = 3$. The other parameters are system size $N = 40 \times 40$ and chemical potential $\mu = 0$.

## D  Dynamics resulting from slow quenches

The main text provides a detailed analysis of the time evolution of the system under a fast quench protocol. It is expected that the results for a sufficiently slow quench would be somehow less interesting as the dynamics would be essentially adiabatic corresponding to a slow, and largely homogeneous, at least initially, growth of the order parameter as temperature is gradually lowered below the critical temperature. For the sake of completeness, in this appendix, we present results for the dynamics, and spatial pattern formation, for a slow quench characterized by $\tau_Q = 500$. We find striking similarities with the fast quench case but also important differences. Results depicted in Figure 14 in the very weak disorder limit $V = 0.001$ show that the observation of time oscillations is greatly delayed with respect to the fast quench

limit but, once they occur, they are qualitatively similar with a fast suppression once the filamentary spatial patterns are fully developed. Interestingly, the crossover from the broken stripe to the fake vortex phase is not clearly observed though it may be due to the limited time scale we have explored numerically or simply due to the weak strength of the random potential. In the presence of a stronger disorder $V = 0.5$, see Figure 15, there is no visible time oscillations. The shape of spatial domains become irregular, and domain walls become thinner over time, but persist within the range of times that we can explore numerically. This rather different, with respect to the fast quench case, spatial structure deserves further exploration.

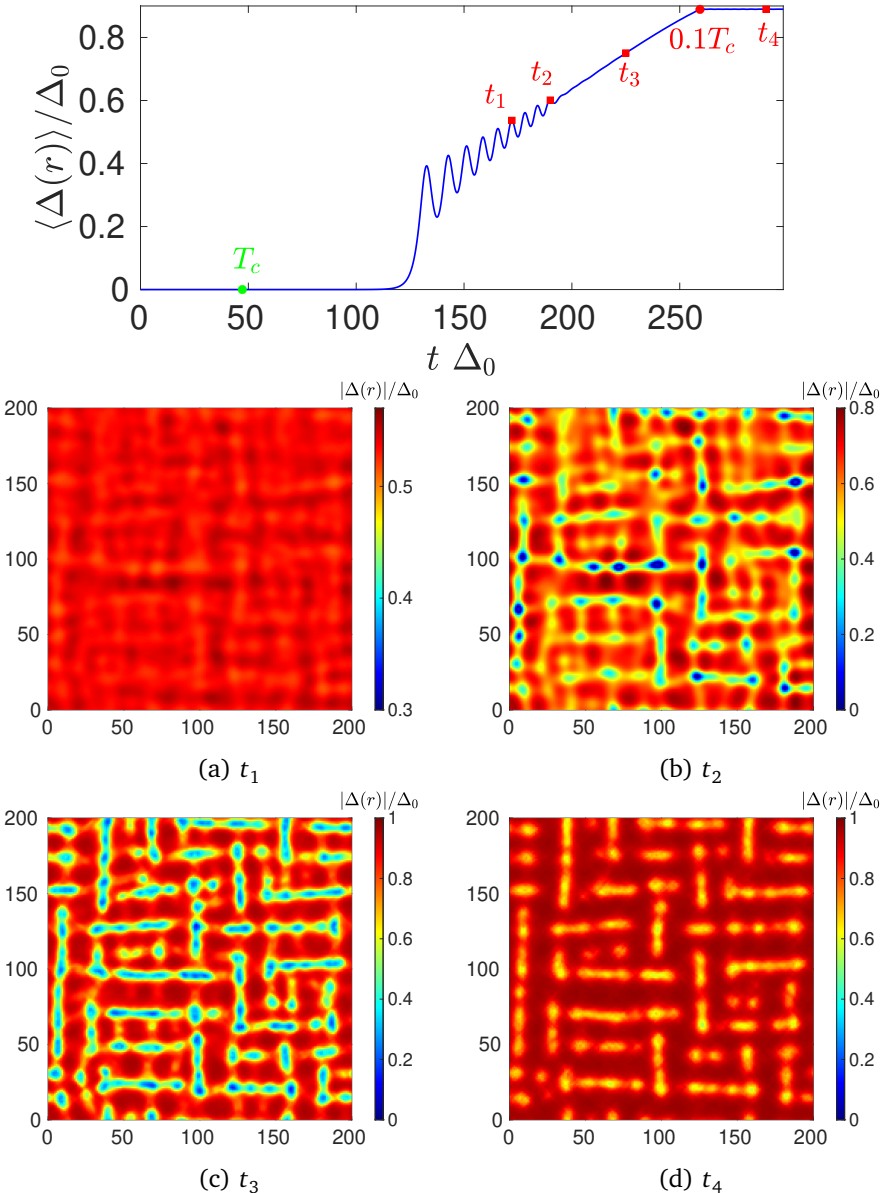

Figure 14: Top: The quenched dynamics of the spatially averaged order parameter $\langle \Delta(\boldsymbol{r}) \rangle$ in the presence of a random potential with disorder strength $V = 0.001$. Bottom: the spatial distribution of the order parameter at times $t_1, t_2, t_3$ and $t_4$ defined in the top plot.

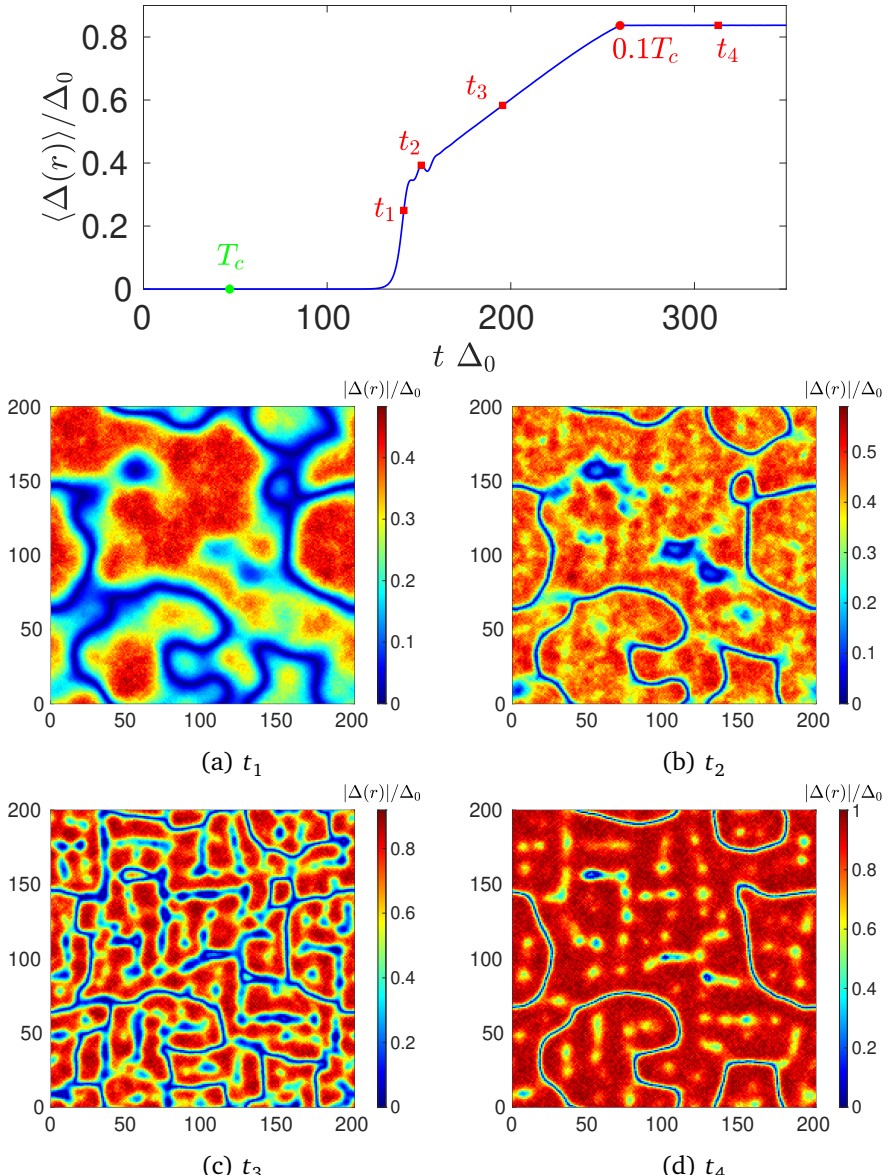

Figure 15: Top: The slow quench dynamics of the spatially averaged order parameter $\langle \Delta(r) \rangle$ in the presence of a weak disorder of strength $V = 0.5$. Bottom: the spatial distribution of the order parameter at times $t_1, t_2, t_3$ and $t_4$ defined in the top plot.

# E    Dynamics in the fast quench, strong coupling limit U = -5

In this appendix, we present results for the quench dynamics in the clean limit ($V = 0$) of the order parameter, using the protocol of the main text (fast quench), in the region of stronger coupling constant $U = -5$. We observe, see Figure 16, qualitatively similar features as for the $U = -3$ case studied in the main text. The spatial averaged order parameter increases exponentially when temperature is below $T_c$, and then exhibits damped oscillations in time. The development of spatial inhomogeneities in the order parameter also followed a similar pattern: the sharp growth of the spatial inhomogeneities occurs around the time in which time oscillations are fully suppressed and a broken stripe phase is followed by the fake vortex phase though the size of these structures is substantially smaller than for $U = -3$. For a more quantitative understanding of this size difference, we study the order parameter correlation

function and the structure factor at time $t_4$ in Figure 16e, when the order parameter is almost at equilibrium, namely, it only experiences very small, non harmonic, oscillations due to the residual collective behavior of Cooper pairs. The results, depicted in Figure 17, show that the order parameter correlation function undergoes oscillations in space. Likewise, the structure factor reveals the existence of a square lattice of fake vortices with typical length $\ell_p \sim 5.6$ in real space. This typical length is much shorter than that for $U = -3$, but still much larger than the superconducting coherence length which, for this value of the coupling, is of the order of the lattice spacing.

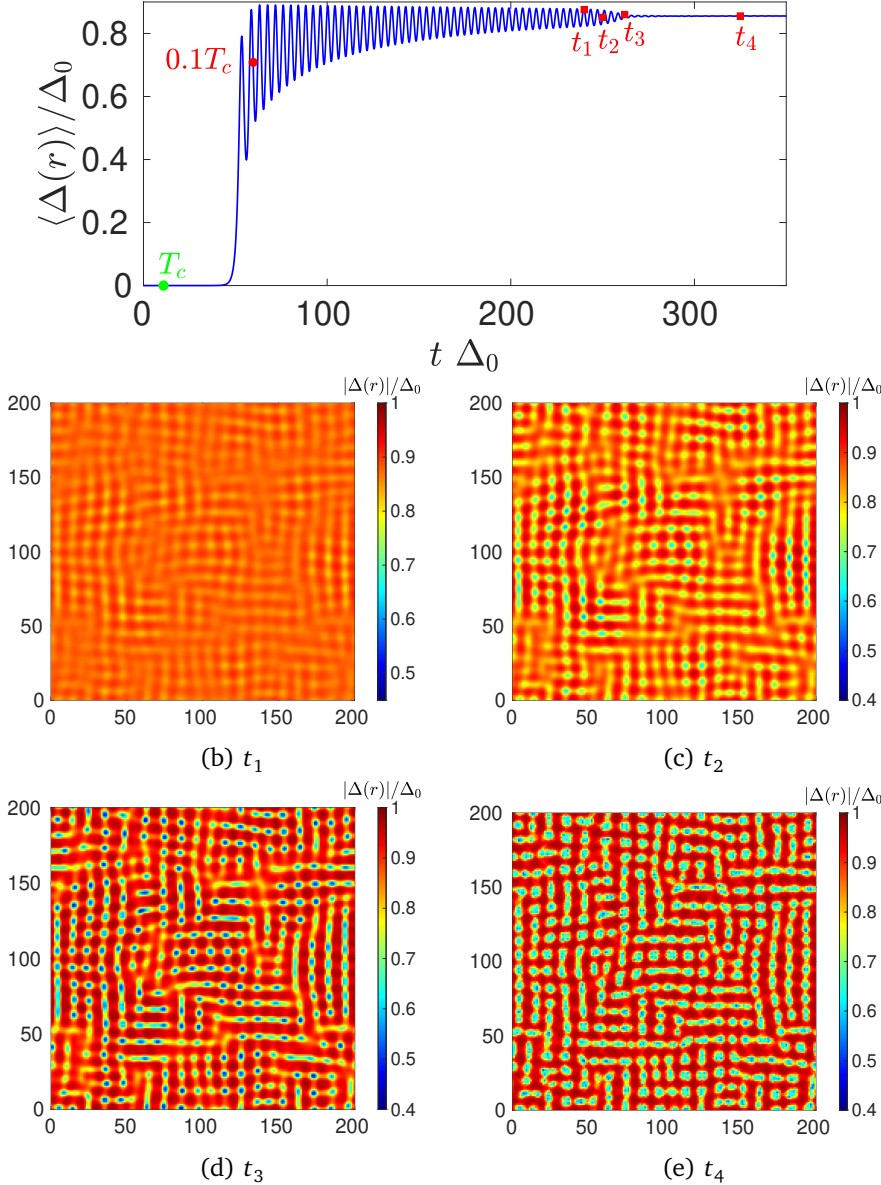

Figure 16: Top: The time evolution of the spatial averaged order parameter $\langle \Delta(\mathbf{r}) \rangle$ in the clean limit, $V = 0$. Bottom; The spatial distribution of the order parameter at the corresponding times $t_1, t_2, t_3$ and $t_4$ shown in the top figure. The system size is $N = 200 \times 200$, the coupling constant $U = -5$ and chemical potential $\mu = -0.45$.

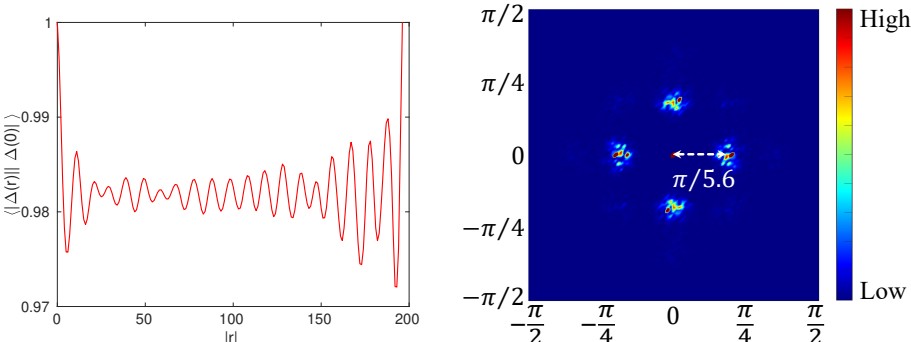

Figure 17: Left: The order parameter correlation function at the equilibrium time, $t_4$ in Figure 16e which is normalized by $\langle \Delta(\boldsymbol{r} = 0)\Delta(0)\rangle$. Right: The structure factor at quasi equilibrium time $t_4$, in Figure 16e. The Bragg's pattern shows a distance to the peaks of around $\pi/5.6$ in momentum space, which corresponds to a typical length of the lattice of about $\ell_p \sim 5.6$ while the superconducting coherence length is much smaller, about the lattice spacing.

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
