# Peer review of "Quenched dynamics and pattern formation in clean and disordered Bogoliubov-de Gennes superconductors"

_SciPost Physics, doi:SciPost Phys. 17, 049 (2024)_

## Round 1 · Referee Report · Anonymous (Referee 1) · 2024-6-18

Strengths

  1. An interesting qualitative prediction related with the formation of position dependent fluctuations of the order parameter after a quench.
  2. Careful comparison of the initial transient behaviour to other approaches.
  3. Careful studying of the effects of disorder
  4. Careful response to referee queries

Weaknesses

  1. The chosen model for the adiabatic time evolution of the electron occupation number with the fast-changing temperature is unphysical, but the authors argue that this has minor consequences.

Report

The authors have satisfactorily answered all my criticism, done the requested checks and implied modifications to the manuscript. I recommend publishing this manuscript in SciPost Physics.

There is one additional remark, which the authors can take into account depending on their choice: this does not affect my decision. Namely, the authors claim that their calculation does not include traces of the collective (phase or amplitude) mode fluctuations of the order parameter, and taking them into account in detail would be far too time-consuming to warrant their investigation. I would tend to disagree with the first statement. In the usual theory of the collective modes, they are obtained as time and position dependent fluctuations around the mean field solution in such a way that the mean field provides the starting point for evaluating that dynamics. An example of such a calculation is given in Phys. Rev. B 99, 224511 (2019). Therefore, considering the dynamics of small corrections to the order parameter is similar to considering the excitation of the collective modes. Therefore, I would claim that the initial oscillatory power-law behaviour is a manifestation of the collective mode dynamics described in PRL 118, 047001 (2017) that I cited in my previous report.

Recommendation

Publish (meets expectations and criteria for this Journal)

---

## Round 1 · Author Response

Dear Editors,

Thanks for forwarding the referee reports. After a careful reading, we have decided to resubmit the SciPost. Both referee reports were constructive and raised interesting issues that have helped to improve the manuscript. We think, we have addressed in detail all referee comments and questions. It has taken some time to prepare the response because we had to obtain new numerical results. For instance, following the referee question about the dependence of the emergent spatial patterns on the nature of the underlying lattice, we obtained results for a triangular lattice that confirmed the referee intuition that it may have an effect. Likewise, in order to answer the question about the dependence on dimensionality, we managed to obtain results for a relatively small 3D lattice that confirmed our argument that two dimensional nature of our setting is not crucial for the observation of spatial inhomogeneities as a consequence of the quenched dynamics. See enclosed a complete list of changes. Finally, we would like to comment on the referee question about long term storage of the companion videos. In case that SciPost offers this possibility, we would be glad to place them in the provided address.

Yours Sincerely,

Fan Bo and Antonio M. Garcia-Garcia

---

## Round 1 · List of Changes

1. We added new Appendix B to introduce the new results of the triangular lattice system, and have mentioned in the abstract, and elsewhere, that the pattern depends on the underlying lattice structure.

  2. We made significant improvement of section II to introduce all the details of our numerical calculations.

  3. The caption of Fig. 1 has been improved. The caption of Fig 2 is also modified accordingly.

  4. We have enlarged Figs. 1, 3, 5

  5. We corrected a typo in the y label of Fig. 4 and replaced “increases with disorder" with "increases with increasing disorder" in the caption.

  6. We corrected the typos found by referee 1 and several others that we found after a careful proofreading.

  7. After eq. (4), we have added detailed explanations, and the physical origin, of the seed that eventually induces large spatial inhomogeneities in the clean limit.

  8. We added a Readme.txt to explain the videos.

  9. In the Discussion section, we have added a justification why our results are not much affected by BKT physics and the Mermin-Wagner theorem.

---

## Editorial Decision

published